# The pooled prevalence of attention-deficit/ hyperactivity disorder among children and adolescents in Ethiopia: A systematic review and meta-analysis

**Desalegn Girma**[1]*, **Zinie Abita**[2], **Amanuel Adugna**[1], **Melsew Setegn Alie**[2], **Nigusie Shifera**[2], **Gossa Fetene Abebe**[1]

1 Department of Midwifery, College of Health Science, Mizan-Tepi University, Mizan-Teferi, Ethiopia,
2 Department of Public Health, College of Health Science, Mizan-Tepi University, Mizan-Teferi, Ethiopia

* desegir@gmail.com

## Abstract

### Background

Attention-deficit/hyperactivity disorder is one of the most common childhood neurobehavioral disorders, which has a serious negative effect on educational achievement, peer relationships, social functioning, behavior, and self-esteem of children. However, the pooled prevalence of attention-deficit/hyperactivity disorder is not well known in Ethiopia. Therefore, the main objective of this systematic review and meta-analysis is to estimate the pooled prevalence of attention-deficit/hyperactivity disorder among children and adolescents in Ethiopia.

### Methods

PubMed, HINARI, Science Direct, Psych INFO, Google Scholar, African Journals Online, and cross-referenced were searched to identify relevant articles. Quality appraisal was done using the Joanna Briggs Institute checklist. Heterogeneity was tested using the I-square statistics. Publication bias was tested using a funnel plot visual inspection. Further, trim and fill analysis was done to correct publication bias. Forest plots and tables were used to present results. The random effect model was used to compute the pooled prevalence of attention-deficit/hyperactivity disorder among children and adolescents.

### Results

The overall pooled prevalence of attention-deficit/hyperactivity disorder among children and adolescents in Ethiopia was 14.2% (95% CI: 8.48, 22.83). Being male (OR: 2.19, 95% CI: 1.54; 3.12), being aged 6–11 years (OR: 3.67, 95% CI: 1.98; 6.83), low family socioeconomic status (OR: 3.45 95% CI: 2.17; 5.47), maternal complication during pregnancy (OR: 3.29, 95% CI: 1.97; 5.51) and family history of mental illness (OR: 3.83, 95% CI:2.17; 6.77) were factors associated with a higher odds of attention-deficit/hyperactivity disorder among children and adolescents.

**Data Availability Statement:** The data used for this study was deposited at Harvard Dataverse

repository URL: https://doi.org/10.7910/DVN/4FYYGN.

**Funding:** The author(s) received no specific funding for this work.

**Competing interests:** The authors have declared that no competing interests exist.

**Abbreviations:** ADHD, Attention-Deficit/Hyperactivity Disorder; JBI, The Joanna Briggs Institute Critical Appraisal Checklist; PRISMA, Preferred Reporting Items for Systematic Review and Meta-Analysis Statement; OR, odds ratio; CI, confidence interval.

## Conclusions

The overall pooled prevalence of attention-deficit/hyperactivity disorder among children and adolescents is high in Ethiopia as compared to previous literature. To reduce the prevalence of attention-deficit/hyperactivity disorder among children and adolescents, emphasis has to be given to prevention, early detection, and management of pregnancy-related complications. Moreover, parents with mental illness should be supported and properly treated to reduce the impact of hostile parenting on their child's health.

## Trial registration

Registered in PROSPERO with ID: CRD42024536334.

## Introductions

Attention-Deficit/Hyperactivity Disorder(ADHD) is one of the most common neurodevelopmental disorders in children [1]. ADHD is usually first identified in school-aged children when it leads to distraction in class [2]. Children with ADHD have trouble focusing, being fidgety, controlling impulsive behaviors [3]. ADHD has three main subtypes: attention deficit, hyperactive-impulsive, and combined types [1, 4]. Children with attention deficit subtype have difficulties paying attention, organizing, or completing tasks [2].On the other hand, children with hyperactive-impulsive subtypes are characterized by high levels of activity, fidgety and difficulty sitting or quiet, waiting their turn, and having trouble with impulsivity [1, 2, 4]. Lastly, children with combined subtypes manifest both attention deficit and hyperactive-impulsive class symptoms [4].

The exact cause of ADHD remains unknown. However, genetic predisposition and environmental factors have been considered as potential risk factors [5–8]. Moreover, other factors such as parent psychopathology also attributed to offspring ADHD. Previous studies reported that the likelihood of ADHD was higher among children whose parents were diagnosed with psychiatric disorders [9, 10], which further reduces the neurocognitive performance of children with ADHD [11]. This implies that children whose parents with psychopathological symptoms should be screened for ADHD and incorporated into treatment programs.

ADHD affects the life of children in many aspects. Such as education is one of the major affected areas; different studies reported that children with ADHD have lower academic achievement than children without ADHD [5, 12–18]. Additionally, Children with ADHD have difficulties in peer relationships, social functioning, and behaving with others [9, 19–26]. Furthermore, studies found that children with ADHD are at higher risk of unintentional accidents [27–33]. Lastly, ADHD can lead children to anxiety and depression [34–36] and poor self-esteem if not appropriately treated [37, 38]. Previous literature reported that ADHD can persist into adulthood [39, 40]. During adulthood, it impairs occupational function and work productivity [41–44], academic achievements [44], family and interpersonal relationships [45] and increases the risk of substance abuse [46]. Moreover, ADHD also increases economic burden on families and society [47, 48].

Globally, the prevalence of ADHD among children and adolescents raged from 5.29% to 8.0% [8, 49–51]. Moreover, according to the systematic review and meta-analysis conducted in the different parts of the continents, the prevalence of ADHD among children and adolescents was found to be 2.9% in Europe [52], 10.90% in America [44], 5.9% in Arab Gulf [53], 10.3%

in the Middle East and North Africa [54] and 5.4% to 8.7% in Africa [55, 56]. This shows that a large number of children and adolescents remain affected by ADHD and are facing difficulties associated with the disorder.

According to studies conducted in different parts of Ethiopia, the prevalence of ADHD among children and adolescents varied across the regions, ranging from the lowest (7.3%) in the Oromia region [57] to the highest (44.2%) in the Amahara region [58]. Moreover, other primary studies have also reported disequilibrium findings regarding the prevalence of ADHD among children and adolescents across regions in Ethiopia, and different factors were identified [59–63]. With this discrepancy, the pooled prevalence of ADHD among children and adolescents has not been estimated in Ethiopia. Therefore, the main purpose of this systematic review and meta-analysis is to estimate the aggregated prevalence of ADHD among children and adolescents and identify its associated factors. The finding of this study could help the policymakers, researchers, program implementers, and other responsible bodies by disclosing the pooled prevalence of ADHD among children and adolescents and identifying its associated factors.

## Methods

### Search strategy

The Preferred Reporting Items for Systematic Review and Meta-Analysis Statement (PRISMA-2020) guideline was used to report the results [64]. International databases such as PubMed, HINARI, Science Direct, Psych INFO, and other sources such as Google Scholar, African Journals Online, and cross reference were searched on March 12, 2024 to obtain relevant studies. The following terms and phrases such as "prevalence", "magnitude", "epidemiology", "attention deficit hyperactivity disorder", "ADHD", "ADDH", "behavioral disorder", "neurodevelopmental disorder" "pediatrics", "children" "under-five children", **"adolescents"** and "Ethiopia" were used to search articles. The Boolean search operators such as "AND" and "OR" were used separately and in combination during database searching **(S1 Table).**

### Eligibility criteria

The inclusion criteria were: 1) original studies that report the prevalence of ADHD among children and adolescents in Ethiopia and/or associated factors, 2) longitudinal studies (using a cross-sectional or case-control design), 3) studies published in English languages and 4) studies available at the electronic source up to March 12, 2024 were incorporated in the study. On the other hand, studies that didn't report the event (i.e. the number of ADHD) or factors, citations without abstract and/or full-text, anonymous reports, editorial reports, and qualitative studies were excluded from the analysis.

### Data extraction

After searching the databases, all the studies were exported to EndnoteX7 to identify and remove duplication. The data was independently extracted using a standardized extraction form by three authors (GF, ZA, and MS). From each study, the author's name, publication year, the number of ADHD, study region, and the predictor of ADHD with odd ratios were extracted.

### Quality assessment/critical appraisal

The Joanna Briggs Institute (JBI) Critical Appraisal Checklist for cross-sectional study design was used to assess the quality of the study [65]. The qualities of the primary studies were

independently assessed by two authors (AA and NS). Any discrepancy between the two authors was handled by taking the mean score of the two authors. The tool has Yes, No, Unclear, and Not Applicable options: "1" is given for "Yes" and "0" is given for other options. The scores were summed and changed to percentages. Finally, 7 studies that received a quality score of >50% were included in this meta-analysis (**S2 Table**).

### Outcome measurement

The prevalence of ADHD among children and **adolescent**s in Ethiopia was the first outcome of this systematic review and meta-analysis. The factors associated with ADHD among children and **adolescent**s were the second outcome of this study. Accordingly, the odd ratio of factors associated with ADHD with its 95% confidence intervals (CI) was extracted from the original studies to compute the pooled odd ratio.

### Statistical analysis

Data entry was done using Microsoft Excel 2013 and then imported into R software version 4.1.3 for further analysis. Data analysis was done using meta-package. Heterogeneity was checked using the I-square test [66]. Heterogeneity was declared as low, medium, and high if the $I^2$ value was 25%, 50%, and 75%, respectively [67]. Subgroup analysis was done using the study region and sex of children and adolescents. To identify the possible source of heterogeneity meta-regression analysis was done using sample size and the publication years. Sensitivity analysis was done by omitting individual studies to detect the contribution of each study to the final pooled prevalence of ADHD. Funnel plot visual inspection was done to identify publication bias. Further, the trim-and-fill imputation was done to correct the publication bias. The forest plot was used to present the prevalence of ADHD with its 95% confidence interval. Tables were also used to present data.The random effect model was used to compute the pooled prevalence of ADHD.

## Results

### Characteristics of included studies

A total of 1,612 studies were searched from PubMed, HINARI, Science Direct, and Psych INFO and other sources such as Google Scholar, African Journals Online, and cross-reference. Among these, 441 studies were from PubMed, 42 studies were from HINARI, 189 studies were from Science Direct, 823 studies were from Psych INFO and the rest 117 studies were searched from other sources. From these studies, 639 studies were excluded due to duplication. From the remaining 973 studies, 955 studies were excluded as not being relevant to the study after reviewing the title and abstract. From the rest 18 studies, the full text of the two studies was not retrieved and removed from the analysis. The rest 16 studies were assessed by reviewing the full text. Finally, a total of 7 studies were eligible and incorporated in the final systematic review and meta-analysis [57–63] (**Fig 1**). All of the studies were conducted using the cross-sectional study design. These studies were done from different parts of Ethiopia (Tigray region, Amhara region, and Oromia region (**Table 1**).

### The pooled prevalence of ADHD among children and adolescents in Ethiopia

A total of seven studies were used to compute the pooled prevalence of ADHD among children and adolescents [57–63]. Accordingly, the pooled prevalence of ADHD among children and adolescents was found to be 14.2% (95% CI: 8.48, 22.83) using the random effect model

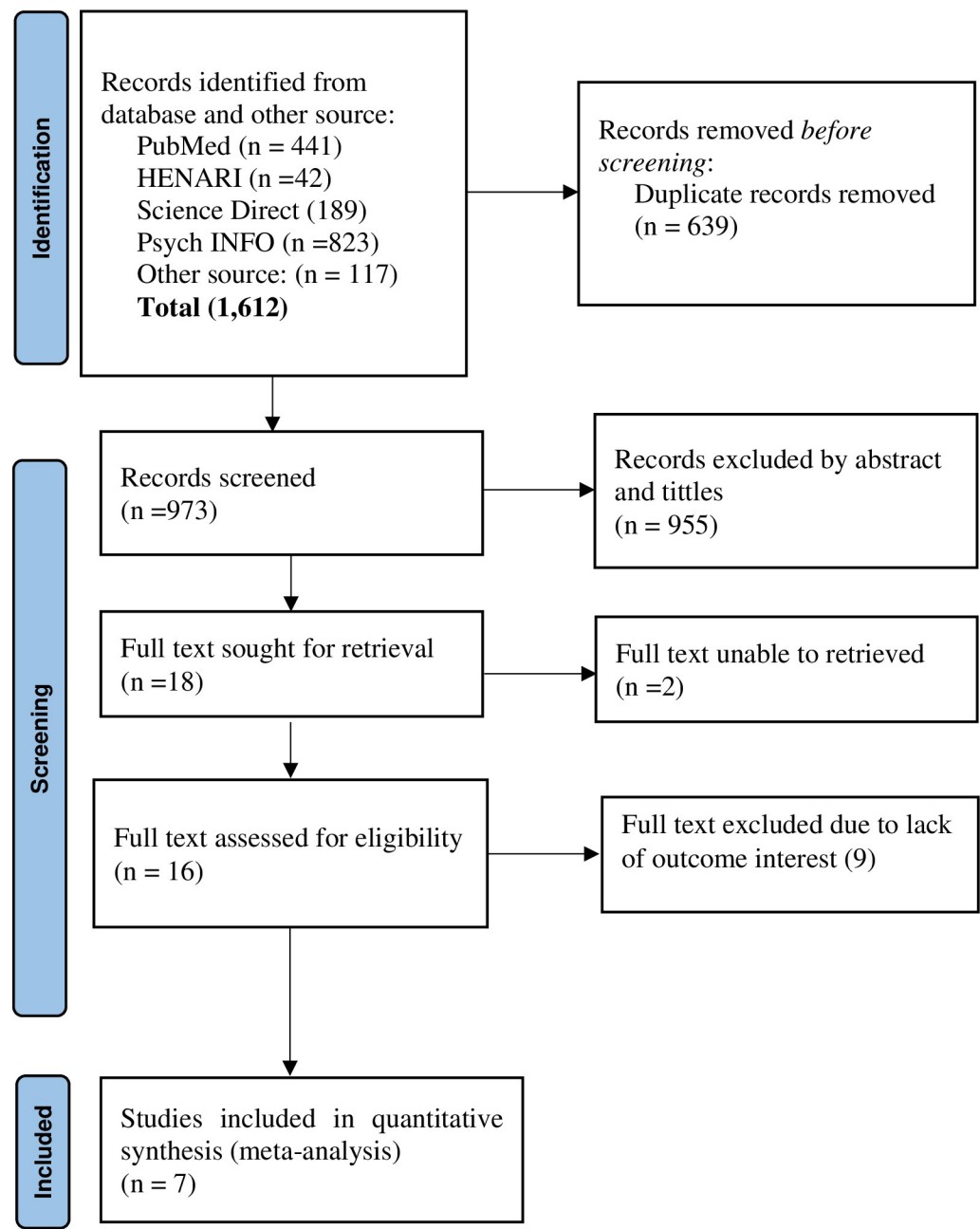

**Fig 1. PRISMA flow chart describing screening protocols of studies for meta-analysis.**

(**Fig 2**). Heterogeneity ($I^2$ = 97%, p-value <0.01) was identified between included studies. Hence, Subgroup analysis was done based on the study regions and sex of children and adolescents. Accordingly, the pooled prevalence of ADHD among males was 9.45% (95% CI: 4.48; 18.85) whereas among females was 4.57% (95% CI: 2.53; 8.14) (**Table 2**). Sensitivity analysis was done to determine the contribution of each study in the final pooled estimate of ADHD. Accordingly, except for one study [58], nearly all studies have equal contributions in the final pooled estimate(**Fig 3**).

**Table 1. Characteristics of studies included in the systematic review and meta-analysis, 2024.**

| Author | Publication year | Regions | Sample size | Number of ADHD | Prevalence (%) |
|---|---|---|---|---|---|
| Lola, et al (2019) [57] | 2019 | Oromia | 1238 | 90 | 7.3 |
| Benti, et al (2021) [59] | 2021 | Oromia | 407 | 34 | 8.4 |
| Mulat, et al(2021) [58] | 2021 | Amhara | 260 | 115 | 44.2 |
| Aliye, et al (2023) [60] | 2023 | Oromia | 504 | 50 | 9.9 |
| Kassa, et al (2018) [61] | 2018 | Tigray | 541 | 100 | 18.5 |
| Tiruneh, et al (2015) [62] | 2015 | Oromia | 387 | 53 | 13.7 |
| Mulu, et al (2021) [63] | 2021 | Amhara | 355 | 46 | 13 |

## Meta-regression

Meta-regression was conducted to identify the possible source heterogeneity of ADHD among children and adolescents using the publication years and sample size. Of these factors, none of them were statistically significant (Table 3).

## Publication bias

Asymmetric distribution was detected in the funnel plot visual inspection (Fig 4). Accordingly, further trim and fill analysis was done to correct publication bias. Accordingly, after adding two studies, the pooled prevalence of ADHD became 18.44% (95% CI: 11.03; 29.19) using the random effect model.

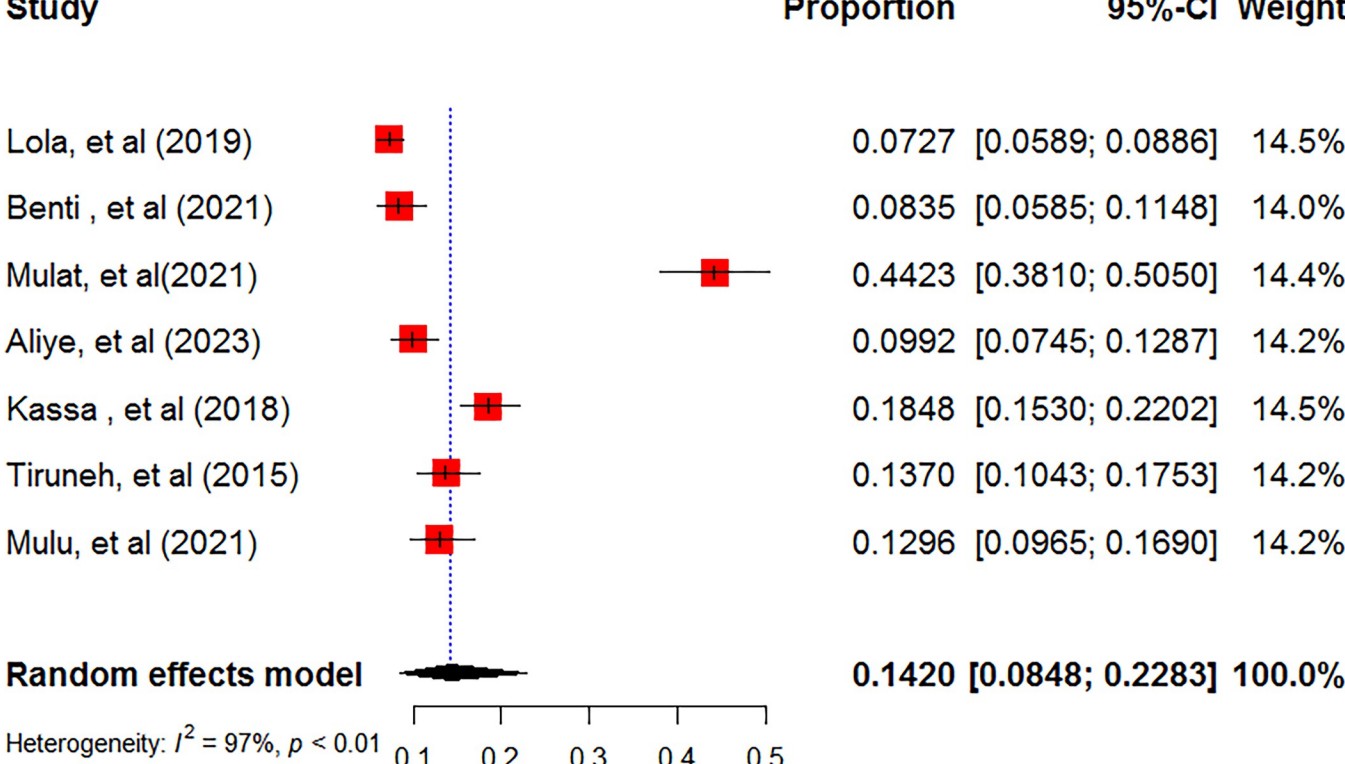

**Fig 2. The forest plots show the pooled prevalence of attention-deficit/hyperactivity disorder among children and adolescents in Ethiopia, 2024.**

**Table 2. Sub-group analysis of ADHD among children and adolescents in Ethiopia, 2024.**

| Variables | Characteristics | Estimated ADHD (95%CI) | I² tests with p-value |
|---|---|---|---|
| Sex | Male | 0.0945 (0.0448; 0.1885) | I² = 97.7%, P-value <0.001 |
| | Female | 0.0457 (0.0253; 0.0814) | I² = 90.5%%, P-value <0.001 |
| Regions | Tigray | 0.1848 [0.1543; 0.2198] | ----- |
| | Amhara | 0.2563 [0.0627; 0.6397] | I² = 98.6%,P-value <0.001 |
| | Oromia | 0.0952 [0.0719; 0.1250] | I² = 80.5%,P-value = 0.001 |

## Factors associated with ADHD among children and adolescents

Seven studies were used to estimate the pooled odd ratio of factors associated with ADHD among children and adolescents [57–63]. Accordingly, the odds of ADHD were 2.19 times (OR: 2.19, 95%CI: 1.54; 3.12) higher among male children and adolescents as compared to female children and adolescents [57–59]. The likelihood of ADHD was 3.67 times (OR: 3.67, 95% CI: 1.98; 6.83) higher among children and adolescents whose ages are 6–11 years as compared to children and adolescents whose ages are greater than 11 years [59, 60]. The odds of having ADHD were 3.45 times (OR: 3.45, 2.17; 5.47) higher among children and adolescents who are from low socio-economic status family members as compared to their counterparts [57, 59, 61]. The odds of having ADHD were 3.29 times (OR: 3.29: 95% CI: 1.97; 5.51) higher

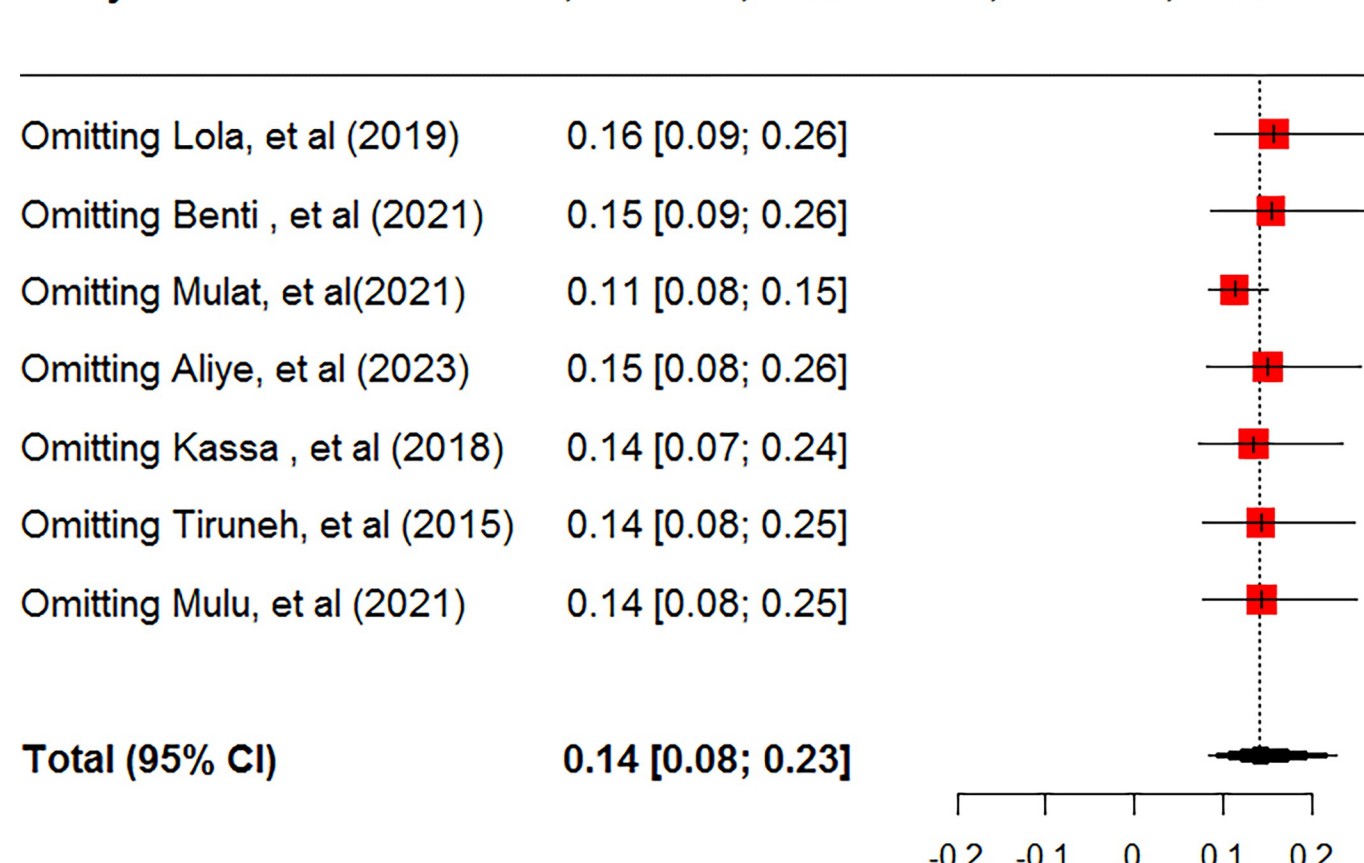

**Fig 3. Sensitivity analysis for pooled prevalence of attention-deficit/hyperactivity disorder among children and adolescents in Ethiopia, 2024.**

**Table 3. Meta-regression analysis using publication years and sample sizes for the possible source of heterogeneity of pooled prevalence of ADHD, Ethiopia, 2024.**

| Variables | Coefficients | P-value |
|---|---|---|
| Publication years | -0.0405 (-0.1412, 0.0603) | 0.99 |
| Sample size | 0.0002 (-0.0019,0.0024) | 0.14 |

among children and adolescents who had maternal complications during pregnancy as compared to their counterparts [59–61]. The likelihood of having ADHD was 3.83 times (OR:3.83, 95% CI: 2.17; 6.77) higher among children and adolescents who have a family history of mental illness as compared with those who didn't have a family history of mental illness[58, 59] (**Table 4**).

## Discussion

To the best of the author's knowledge, there is no pooled estimate of ADHD among children and adolescents in Ethiopia. Therefore, this systematic review and meta-analysis provide the pooled prevalence of ADHD among children and adolescents. Accordingly, the overall pooled prevalence of ADHD among children and adolescents was found to be 14.2% (95% CI: 8.48, 22.83) using the random effect model. The finding is higher than the systematic review and meta-analysis studies conducted elsewhere [8, 43, 49–55]. The possible elucidation for the discrepancy might be the difference in culture and socioeconomic characteristics of study participants that are consistently associated with increased rates of ADHD in children and adolescents [68, 69]

In this systematic review and meta-analysis, the odds of having ADHD are higher among males than females. The finding is synonymous with studies conducted elsewhere [39, 70, 71]. The possible clarification might be due to differences in the expression of the disorder between males and females [72]. Most of the time females present with internalized symptoms (such as

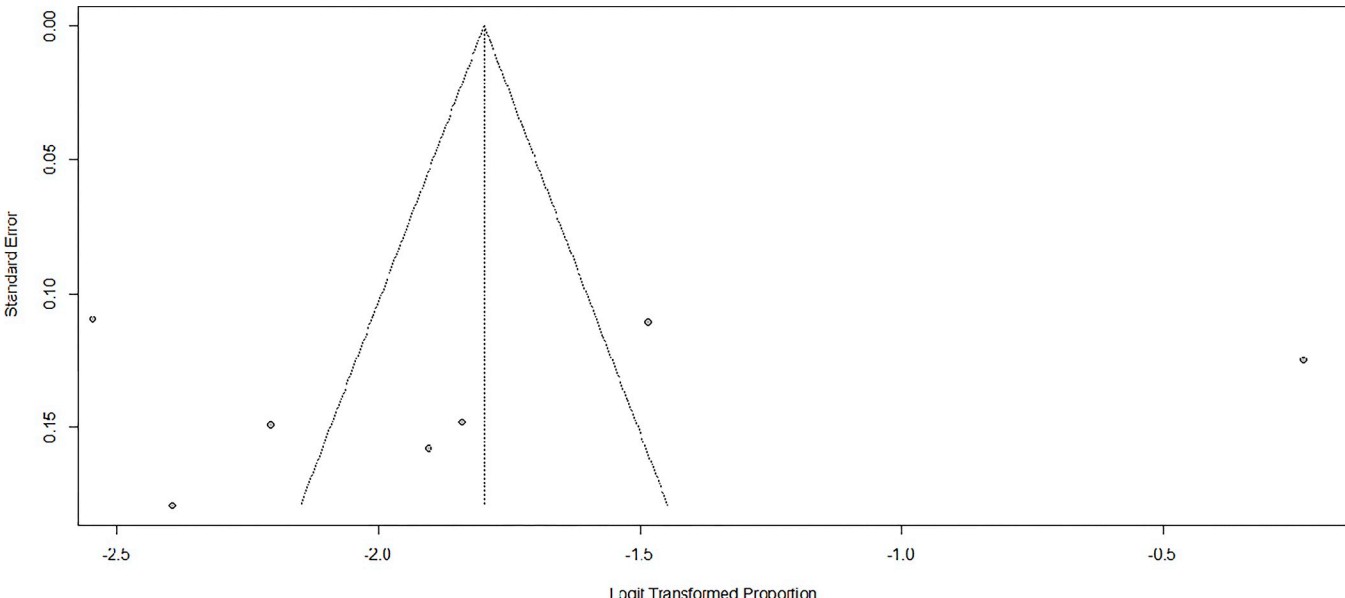

**Fig 4. Funnel plot showing publication bias among studies used to compute the pooled prevalence of attention-deficit/hyperactivity disorder among children and adolescents in Ethiopia, 2024.**

**Table 4. Factors associated with ADHD among children and adolescents, Ethiopia, 2024.**

| Variables | Included studies | OR (95% CI) | Pooled OR (95% CI) | Heterogeneity |
|---|---|---|---|---|
| Male sex | Lola, et al (2019 | 1.81 (1.13; 2.91) | 2.19(1.54; 3.12) | $I^2$ = 0%, p-value = 0.47 |
| | Benti, et al (2021) | 3.07 (1.16; 9.39) | | |
| | Mulat, et al(2021) | 2.70 (1.46; 4.97) | | |
| Age 6–11 years | Benti, et al (2021) | 3.38 (1.22; 9.39) | 3.67(1.98; 6.83) | $I^2$ = 0%, p-value = 0.83 |
| | Aliye, et al (2023) | 3.86 (1.7700; 8.43) | | |
| low family socioeconomic status | Lola, et al (2019) | 2.43 (1.28; 4.58) | 3.45 (2.17; 5.47) | $I^2$ = 38.9%, p-value = 0.19 |
| | Benti, et al (2021) | 3.47 (1.25; 10.1) | | |
| | Kassa, et al (2018) | 6.55 (2.76; 15.57) | | |
| Maternal complications during pregnancy | Benti, et al (2021) | 3.05 (1.10; 8.48) | 3.29 (1.97; 5.51) | $I^2$ = 0%, p-value = 0.97 |
| | Aliye, et al (2023) | 3.56 (1.44; 8.79) | | |
| | Kassa, et al (2018) | 3.26 (1.49; 7.15) | | |
| Family history of mental illness | Benti, et al (2021) | 3.60 (1.78; 8.21) | 3.83 (2.17; 6.77) | $I^2$ = 0%, p-value = 0.81 |
| | Mulat, et al(2021) | 4.14(1.76; 9.68) | | |
| Preterm | Kassa, et al (2018 | 1.87 (1.01 3.49) | 4.17(0.7454; 23.35) | $I^2$ = 84.5%,p-value = 0.01 |
| | Tiruneh, et al (2015) | 10.92(3.253;36.67) | | |

OR: odds ratio. CI: confidence interval.

anxiety, depression, and low self-esteem) and missed from diagnosis [73, 74]. Whereas males with ADHD externalize the symptoms (such as hyperactive or impulsive and aggressive behavior) than females and can be easily diagnosed [75]. This may make the difference between male and female.

In this systematic review and meta-analysis, the likelihood of having ADHD is higher among children and adolescents whose ages are 6–11 years as compared to children and adolescents whose ages are greater than 12 years. The finding is in agreement with the study conducted elsewhere [39]. The possible justification might be due to the reduction in hyperactive-impulsive symptoms, and stability of inattentive symptoms as age increased.

The other observed finding of this study is that the odds of having ADHD are higher among children and adolescents who are from low socio-economic status family members as compared to their counterparts. The finding is supported by the study conducted elsewhere [69, 76, 77]. The possible elucidation might be family with low socioeconomic status may not meet the needs of their children such as school choice and others. This can change the behavior of children and adolescents. Moreover, low socioeconomic status increases the risk of pregnancy-related complications, which is the known risk factor for ADHD [39, 78].

The odds of having ADHD are higher among children and adolescents who have maternal complications during pregnancy as compared to their counterparts. The finding is supported by studies conducted elsewhere [39, 78, 79]. The possible explanation might be maternal complications during pregnancy might cause multi-organ damage including the central nervous system, which can affect fetal neurodevelopment and may predispose to ADHD [80, 81].

This systematic review and meta-analysis revealed that the odds of having ADHD are higher among children and adolescents who have a family history of mental illness than those who don't have a family history of mental illness. The finding is consistent with studies conducted elsewhere [70, 82–85]. This is the fact that genetic predisposition contributes to offspring ADHD [86, 87]. Moreover, this is the fact that hostile parent interaction can affect their child's behaviors and increase the risk of being diagnosed with ADHD [88]. Generally, more emphasis has to be given to the prevention and treatment of children and adolescents with

ADHD. Moreover, researchers, program implementers, and policymakers should consider the aforementioned factors in their strategic plans.

## Limitations

This systematic review and meta-analysis has the following limitations: In this analysis, articles published only in English were included, only three regions were included in the analysis which some regions may not be represented in the study and some factors associated with ADHD were excluded from the analysis because of there were reported only in one primary article and/or classified as different way from the included articles. Moreover, though we rigorously searched for studies conducted on treatment modalities of ADHD among children and adolescents in Ethiopia, we can't access any published study on treatment modalities of ADHD in Ethiopia. Such that, we can't review the primary treatment modalities of ADHD among children and adolescents in Ethiopia.

## Conclusions

The overall pooled prevalence of ADHD among children and adolescents is high in Ethiopia as compared to previous literature. Being male, having low family socioeconomic status, being aged 6–11 years, having maternal complications during pregnancy, and family history of mental illness are factors associated with higher odds of ADHD among children and adolescents. To reduce the prevalence of ADHD among children, emphasis has to be given to prevention, early detection, and management of pregnancy-related complications. Moreover, parents with mental illness should be supported and properly treated to reduce the impact of hostile parenting on their child's health. Furthermore, in Ethiopia, further studies on the treatment modality of ADHD and its effects have to be done to enhance the quality of life of children and adolescents with ADHD.

## Supporting information

**S1 Checklist. PRISMA 2020 checklist.**
(DOCX)

**S1 Table. Search terms summary for the pooled prevalence of attention-deficit/hyperactivity among children and adolescents, Ethiopia, 2024.**
(DOCX)

**S2 Table. Critical appraisal of studies included in the systematic review and meta-analysis for pooled prevalence of attention-deficit/hyperactivity disorder among children and adolescents in Ethiopia, 2024.**
(DOCX)

## Author Contributions

**Conceptualization:** Desalegn Girma.

**Data curation:** Desalegn Girma, Zinie Abita, Amanuel Adugna, Melsew Setegn Alie, Nigusie Shifera, Gossa Fetene Abebe.

**Formal analysis:** Desalegn Girma, Gossa Fetene Abebe.

**Investigation:** Desalegn Girma.

**Methodology:** Desalegn Girma, Zinie Abita, Amanuel Adugna, Melsew Setegn Alie, Nigusie Shifera, Gossa Fetene Abebe.

**Software:** Desalegn Girma, Amanuel Adugna, Melsew Setegn Alie, Nigusie Shifera, Gossa Fetene Abebe.

**Writing – original draft:** Desalegn Girma, Zinie Abita.

**Writing – review & editing:** Desalegn Girma, Amanuel Adugna, Melsew Setegn Alie, Nigusie Shifera, Gossa Fetene Abebe.

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
