## [Decision Letter · Decision Letter 0]

22 May 2024

PONE-D-24-12742The pooled prevalence of attention-deficit/hyperactivity disorder among children and adolescents in Ethiopia: A Systematic Review and Meta-Analysis.PLOS ONE

Dear Dr. Girma,

Thank you for submitting your manuscript to PLOS ONE. After careful consideration, we feel that it has merit but does not fully meet PLOS ONE’s publication criteria as it currently stands. Therefore, we invite you to submit a revised version of the manuscript that addresses the points raised during the review process. Both Reviewers recognized the value of your work. Please refer to their comments to improve its quality, especially the ones related to methodology and adherence to Plos One guidelines.

We look forward to receiving your revised manuscript.

Kind regards,

Simone Varrasi

Academic Editor

PLOS ONE

Journal Requirements:

2. In the online submission form, you indicated that [The data is available at the corresponding author and can be provided upon request.]. 

Reviewers' comments:

Reviewer's Responses to Questions

**Comments to the Author**

1. Is the manuscript technically sound, and do the data support the conclusions?

Reviewer #1: Partly

Reviewer #2: Yes

2. Has the statistical analysis been performed appropriately and rigorously? 

Reviewer #1: Yes

Reviewer #2: Yes

3. Have the authors made all data underlying the findings in their manuscript fully available?

Reviewer #1: No

Reviewer #2: Yes

4. Is the manuscript presented in an intelligible fashion and written in standard English?

Reviewer #1: Yes

Reviewer #2: Yes

5. Review Comments to the Author

Reviewer #1: Dear authors,

your systematic review and meta-analysis is interesting dealing with gathering evidence on ADHD in Ethiopia.

Some amendments are necessary in order to improve the overall quality of the manuscript:

1. Into the introduction, some information about ADHD in adulthood are interesting to take into account. For this reason, you could take some information by https://doi.org/10.3390/educsci13010037;

2.The manuscript demonstrates strong writing and methodological validity, although it's worth noting that the meta-analysis was based on only 7 studies, which may be seen as a limitation. Additionally, the evidence presented suggests no significant variations in ADHD prevalence between the Ethiopian population and global trends, particularly in terms of gender disparities or exacerbating factors. Consequently, the study doesn't significantly contribute to the existing scientific discourse.

To enhance the originality of the study, one suggestion is to incorporate a section that systematically reviews or narratively explores the primary treatment modalities available for Ethiopian children and adolescents with ADHD. This addition could introduce a novel perspective and broaden the scope of the research, potentially offering valuable insights into diverse cultural contexts and treatment approaches.

Reviewer #2: Dear authors,

thank you for submitting your work to the journal PLOS ONE and for focusing your study on the topic of ADHD, which is increasingly relevant and at the center of scientific debate. The work sheds important light on the prevalence and associated factors of ADHD among children and adolescents in Ethiopia.

Thank you for your contribution to this essential field of research.

However, some of the steps described within this project need further clarification.

In particular, the following observations could go a long way towards clarifying certain aspects that need to be revised and reformulated.

1. The references section should be formatted according to the PLOS ONE guidelines, please refer to https://journals.plos.org/plosone/s/submission-guidelines .

2. The denomination and description of the table should be placed below it, according to the guidelines.

3. Given the lack of information due to various reasons in Ethiopia and the need to demonstrate to policymakers, researchers, program implementers, and other responsible bodies the importance of understanding the pooled prevalence of ADHD among children and adolescents and identifying its associated factors, it is crucial to highlight the future perspectives for these children as they transition into adulthood, in order to remark the impact of this clinical condition. For more detailed insights, please refer to https://doi.org/10.3390/educsci13010037.

4. I would also suggest including within the introductory section the issue regarding parents with psychopathology in a more cohesive manner, as this topic is mentioned only in the abstract and conclusions without being well integrated into the context with specific references (e.g. 46-47; 268-269).

6. PLOS authors have the option to publish the peer review history of their article (what does this mean?). If published, this will include your full peer review and any attached files.

Reviewer #1: No

Reviewer #2: No

---

## [Author Response · Author response to Decision Letter 0]

23 Jun 2024

Reviewer #1 comments and an author response

Your systematic review and meta-analysis is interesting dealing with gathering evidence on ADHD in Ethiopia

Authors: I would like to express my heart-felt thanks for your devoted time to review our manuscript

Reviewer comment #1: Into the introduction, some information about ADHD in adulthood are interesting to take into account. For this reason, you could take some information by https://doi.org/10.3390/educsci13010037;:

Authors Response: dear reviewer, we thank you heartily for your constructive comments. As per your suggestion we incorporated in the revised manuscript, please, look on page 5 from lines numbers 86-89

Comment #1: The manuscript demonstrates strong writing and methodological validity, although it's worth noting that the meta-analysis was based on only 7 studies, which may be seen as a limitation. Additionally, the evidence presented suggests no significant variations in ADHD prevalence between the Ethiopian population and global trends, particularly in terms of gender disparities or exacerbating factors. Consequently, the study doesn't significantly contribute to the existing scientific discourse.

To enhance the originality of the study, one suggestion is to incorporate a section that systematically reviews or narratively explores the primary treatment modalities available for Ethiopian children and adolescents with ADHD. This addition could introduce a novel perspective and broaden the scope of the research, potentially offering valuable insights into diverse cultural contexts and treatment approaches.

Authors Response: Thanks very much for your comments and concerns. We accepted your comments. Dear, reviewer, we searched as much as our best for studies conducted on treatment modalities of ADHD among children and adolescents in Ethiopia, however, we can’t access any published study or literature on treatment modalities of ADHD in Ethiopia. Such that, we can’t review the primary treatment modalities of ADHD among children and adolescents in Ethiopia. Dear reviewer, in this regard, we considered your comments as a major limitation of our study and we incorporated under our limitation (please look on page from lines numbers 290-293). Moreover, we recommended further studies on the treatment modality of ADHD and its effects among children and adolescents with ADHD (please look, on page 15 from lines numbers 302-303).

Dear reviewer, this indicates that ADHD is negated in Ethiopia. Dear reviewer, in our study, at least, we disclosed the burden of ADHD among children in Ethiopia, which can alarm the governments and urge the policymaker to endorse strategies or programs, and treatment guidelines related to ADHD to improve the quality of children with ADHD and to prevent the Lifespan impact of ADHD. In this regard, this study may have importance in a country, which has limited evidence regarding ADHD.

Dear reviewer, having this, we are ready to consider your further recommendation for any reservation do you have.

Response to Reviewer #2:

Reviewer: thank you for submitting your work to the journal PLOS ONE and for focusing your study on the topic of ADHD, which is increasingly relevant and at the center of scientific debate. The work sheds important light on the prevalence and associated factors of ADHD among children and adolescents in Ethiopia.

Thank you for your contribution to this essential field of research.

However, some of the steps described within this project need further clarification.

In particular, the following observations could go a long way towards clarifying certain aspects that need to be revised and reformulated.

Authors: I would like to express my heart-felt thanks for your devoted time to review our manuscript

Comment 1: The references section should be formatted according to the PLOS ONE guidelines, please refer to https://journals.plos.org/plosone/s/submission-guidelines .

Authors Response: Thanks very much, for your concern and comments, dear reviewer, amendment was done based on the submission-guidelines on reference No.1 and 2

2. The denomination and description of the table should be placed below it, according to the guidelines.

Authors Response: Thanks very much, for your concern and comments. Dear reviewer, descriptions were incorporated in the revised manuscript for supplemental tables and figures, which are not uploaded as supplemental material or uploaded separately as figures.

Comment 2: Given the lack of information due to various reasons in Ethiopia and the need to demonstrate to policymakers, researchers, program implementers, and other responsible bodies the importance of understanding the pooled prevalence of ADHD among children and adolescents and identifying its associated factors, it is crucial to highlight the future perspectives for these children as they transition into adulthood, in order to remark the impact of this clinical condition. For more detailed insights, please refer to https://doi.org/10.3390/educsci13010037.

Authors Response: dear reviewer, we thank you heartily for your constructive comments. As per your suggestion, we incorporated it in the revised manuscript, please, look on page 5 from lines numbers 86-89

 Comment 3: I would also suggest including within the introductory section the issue regarding parents with psychopathology in a more cohesive manner, as this topic is mentioned only in the abstract and conclusions without being well integrated into the context with specific references (e.g. 46-47; 268-269).

Authors Response: dear reviewer, we thank you heartily for your constructive comments. As per your suggestion, we incorporated it in the revised manuscript, please, look on page 4 from lines numbers 73-79

---

## [Decision Letter · Decision Letter 1]

24 Jun 2024

PONE-D-24-12742R1The pooled prevalence of attention-deficit/hyperactivity disorder among children and adolescents in Ethiopia: A Systematic Review and Meta-Analysis.PLOS ONE

Dear Dr. Girma,

Thank you for submitting your manuscript to PLOS ONE. After careful consideration, we feel that it has merit but does not fully meet PLOS ONE’s publication criteria as it currently stands. Therefore, we invite you to submit a revised version of the manuscript that addresses the points raised during the review process.

We look forward to receiving your revised manuscript.

Kind regards,

Simone Varrasi

Academic Editor

PLOS ONE

Journal Requirements:

Reviewers' comments:

Reviewer's Responses to Questions

**Comments to the Author**

1. If the authors have adequately addressed your comments raised in a previous round of review and you feel that this manuscript is now acceptable for publication, you may indicate that here to bypass the “Comments to the Author” section, enter your conflict of interest statement in the “Confidential to Editor” section, and submit your "Accept" recommendation.

Reviewer #1: (No Response)

Reviewer #2: (No Response)

2. Is the manuscript technically sound, and do the data support the conclusions?

Reviewer #1: Yes

Reviewer #2: Yes

3. Has the statistical analysis been performed appropriately and rigorously? 

Reviewer #1: Yes

Reviewer #2: N/A

4. Have the authors made all data underlying the findings in their manuscript fully available?

Reviewer #1: No

Reviewer #2: Yes

5. Is the manuscript presented in an intelligible fashion and written in standard English?

Reviewer #1: Yes

Reviewer #2: Yes

6. Review Comments to the Author

Reviewer #1: Dear authors,

thank you for the revisions made. However, please review the form of the references, as it may not be correct for the journal. It is suggested to use reference software (e.g. Zotero, Mendeley).

Reviewer #2: Thank you for reviewing your work.

In my opinion, the manuscript aligns with the journal's standards.

7. PLOS authors have the option to publish the peer review history of their article (what does this mean?). If published, this will include your full peer review and any attached files.

Reviewer #1: No

Reviewer #2: No

---

## [Author Response · Author response to Decision Letter 1]

26 Jun 2024

Reviewer #1: Dear authors,

thank you for the revisions made. However, please review the form of the references, as it may not be correct for the journal. It is suggested to use reference software (e.g. Zotero, Mendeley).

Authors response: Dear reviewer thank you for comments and concern, we used EndNote software which is one of the popular citation manager as like Zotero, Mendeley. We have no experience on Zotero, Mendeley. Dear reviewer, we have article that published in Plos One and other journal which was cited using EndNote as like as our current manuscript.

Reviewer #2: Thank you for reviewing your work.

In my opinion, the manuscript aligns with the journal's standards.

Authors response: Dear reviewer thank you very much for your devoted time to review our manuscript.

---

## [Editor Report · Decision Letter 2]

2 Jul 2024

The pooled prevalence of attention-deficit/hyperactivity disorder among children and adolescents in Ethiopia: A Systematic Review and Meta-Analysis.

PONE-D-24-12742R2

Dear Dr. Girma,

We’re pleased to inform you that your manuscript has been judged scientifically suitable for publication and will be formally accepted for publication once it meets all outstanding technical requirements.

Kind regards,

Simone Varrasi

Academic Editor

PLOS ONE
---

## [Editor Report · Acceptance letter]

8 Jul 2024

PONE-D-24-12742R2 

PLOS ONE

Dear Dr. Girma, 

I'm pleased to inform you that your manuscript has been deemed suitable for publication in PLOS ONE. Congratulations! Your manuscript is now being handed over to our production team.

Kind regards, 

on behalf of

Dr. Simone Varrasi 

Academic Editor

PLOS ONE